# Limitation of Futile Therapy in the Opinion of Nursing Staff Employed in Polish Hospitals—Results of a Cross-Sectional Study

**DOI:** 10.3390/ijerph192416975

**Published:** 2022-12-17

**Authors:** Maria Damps, Maksymilian Gajda, Malgorzata Kowalska, Ewa Kucewicz-Czech

**Affiliations:** 1Department of Anaesthesiology and Intensive Care, Upper Silesian Child Health Centre, Faculty of Medical Sciences in Katowice, Medical University of Silesia, Medyków 16, 40-752 Katowice, Poland; 2Department of Epidemiology, Faculty of Medical Sciences in Katowice, Medical University of Silesia, 40-752 Katowice, Poland; 3Faculty of Medical Sciences in Katowice, Medical University of Silesia, 40-752 Katowice, Poland

**Keywords:** limitation of futile therapy, end-of-life care, opinion of nursing staff

## Abstract

The debate on limiting futile therapy in the aspect of End of Life (EoL) care has been going on in Poland over the last decade. The growing demand for EoL care resulting from the aging of societies corresponds to the expectation of a satisfactory quality of life and self-determination. The authors designed a cross-sectional study using a newly designed questionnaire to assess the opinions of 190 nurses employed in intensive care units (ICUs) on futile therapy, practices, and the respondents’ approach to the issue. The problem of futile therapy and its clinical implications are known to the nursing community. Among the most common reasons for undertaking futile therapy in adult patients, the respondents declared fear of legal liability for not taking such actions (71.58%), as well as fear of being accused of unethical conduct (56.32%), and fear of talking to the patient/patient’s family and their reaction (43.16%). In the case of adult patients, the respondents believed that discontinuation of futile therapy should be decided by the patient (84.21%), followed by a doctor (64.21%). As for paediatric patients, two-thirds of the respondents mentioned a doctor and a court (64.74% and 64.21%, respectively). Overall, 65.26% of the respondents believe and agree that the comfort of the patient’s last days is more important than the persistent continuation of therapy and prolonging life at all costs. The presented results clearly show the attitude of the respondents who defend the patient’s dignity and autonomy.

## 1. Introduction

In the latest issue of the Surviving Sepsis Campaign guidelines [1] a new section ‘Long-term outcomes and goals of care’ has been added. The authors of the guidelines, including patient representatives, wanted to agree on the goals and boundaries of care to determine when to move toward palliative care. An increasingly asked question is whether we want invasive treatments if they do not increase the chance of survival. The measure of medical success is no longer the mere fact of survival, but an acceptable quality of life after treatment [2]. As a satisfactory quality of life is perceived individually and subjectively by each patient, both doctors and patients begin to doubt the advisability of futile therapy. In the opinion of an increasing number of clinicians, especially specialists in anaesthesiology and intensive care, the medicalization of death raises doubts as to the advisability of futile medical care [3,4]. The concept of futile therapy is not a new term in clinical practice but in Poland, attention has only recently been paid to the ethical, clinical, and legal dilemmas of this issue. We have guidelines for the discontinuation of therapy in both adults and children [5,6]. However, this problem evokes strong emotions, sometimes misunderstanding, and above all, requires extensive discussion in the medical community. The authors designed a cross-sectional study that used a newly designed questionnaire to assess the opinions of nurses and doctors employed in intensive care units (ICUs) on futile therapy, practices, and the respondents’ approach to the issue. This paper presents the results of the study in which the opinions of nurses on futile therapy were assessed, along with determinants related to socio-demographic factors and the type of employment.

## 2. Material and Methods

In 2020, a cross-sectional study was conducted using the proprietary questionnaire. The Polish-language questionnaire consisted of 2 parts with a total of 28 questions. The first part consisted of 10 questions and concerned socio-demographic characteristics such as gender, age, and religion, but also professional work, including practice and workplace. The second part, which consisted of 18 questions, concerned the main topic, i.e., the respondents’ opinions on various aspects of futile therapy (including the reasons for its implementation or the legitimacy of its limitation). The questionnaire was in a traditional paper form and contained both open and closed questions (single and multiple choice); the respondents were asked to answer questions on their own. The questions were prepared by the authors of the study after consultation with Polish experts in the discussed field. The questionnaire was validated in a group of 20 randomly selected medical workers, who were surveyed twice a week with the use of the same tool. The percentage compliance of the answers in the test and retest was assessed, and Cohen’s kappa for the key questions was determined. The obtained results indicate good or very good repeatability of the questions. Nursing staff employed in hospitals located in the Silesian Voivodeship participated in the first stage of the study. Participation was fully voluntary and anonymous, and it did not involve any additional costs for the respondents. The participants did not receive any gratuity. The study design was approved by the Ethics Committee of the Medical University of Silesia (KNW/0022/KB/284/18). In total, 250 nurses were invited to the study, which constituted over 1% of professionally active nurses employed in hospitals of the Silesian Voivodeship.

### Statistical Analysis

The collected data were subjected to statistical analysis in which the classic methods of descriptive and analytical statistics were used. Qualitative variables were presented as numbers and percentage values, and the assessment of intergroup differences was presented using the chi-squared test. The Shapiro–Wilk test was used to describe the distribution of quantitative variables. Due to the distribution of variables deviating from the normal distribution, non-parametric tests (e.g., the Mann–Whitney *U* test) were used in the analysis of intergroup differences, and the median with the interquartile range (IQR) was used for the description of the variables. When interpreting the results, the criterion of statistical significance *p* < 0.05 was adopted. As stated earlier, Cohen’s kappa statistic was used to assess the repeatability of answers (test–retest) in the group of 20 respondents (median age 51 years, 9 females). For the interpretation of the results, we used the classical criteria proposed by Landis and Koch [7]. The repeatability of answers for practically all questions turned out to be good or very good (Cohen’s Kappa values of 0.61 and above), legitimizing the results and the conclusions drawn on their basis. Additionally, the item cluster analysis was conducted with the iclust function (available in the “psych” package within R statistical software, version 4.1.0). It allowed us to confirm the reliability of the research tool (alpha values of the main 3 clusters in the range of 0.83 to 0.88).

Detailed data are presented in the results section. All calculations were performed using the Statistica 13.3 package-StatSoft Polska, TIBCO Gold Partner (except the reliability assessment).

## 3. Results

Out of 250 people invited to participate, 190 nurses joined the study (the participation rate was 76%). The detailed characteristics of the respondents are presented in Table 1. The median age was 43 years (IQR = 18, range 23–60), with the majority of women (81.58%), who were significantly older than men (median age 43 vs. 32.5 years, respectively; *p* = 0.02) and had significantly longer work experience (20 vs. 7.5 years, respectively; *p* = 0.01). Catholics constituted the majority (60.00%), and only 28.95% of the respondents declared weekly attendance at a service in a Catholic church.

### 3.1. Work Experience and Workplace

The median work experience was 19 years (IQR = 22, range 0.2–40 years), although the most numerous group included people working for no more than five years. The vast majority of nurses declared higher education, with more than half (58.95%) being at the bachelor’s level and every fifth person holding a master’s degree (22.11%). Most of the respondents (63.16%) worked in the intensive care units (ICUs); 41.05% of nurses worked with paediatric patients. It is worth mentioning that the majority of respondents were employed in teaching hospitals with ICUs (61.58% of respondents).

### 3.2. Experience and Opinions Related to Futile Therapy

Questions about the experience and opinions of the surveyed nurses on futile therapy were important for the study; detailed results regarding the respondents’ declarations are presented in Table 2. An important question concerned the specificity of professional work in terms of possible contact with dying patients. More than half of the respondents (60.53%) declared that such a possibility occurred quite often, while the vast majority of the respondents (81.58%) witnessed the patient’s illness and death. Three out of four respondents (approx. 74.74%) experienced futile therapy while working with adult patients, while fewer than every third person (30.00%) had such experience in paediatric practice. One in three respondents declared adherence to the futile therapy cessation protocol in adult ICU patients and in the event of an unfavourable prognosis (37.74% and 37.89%, respectively). The majority of respondents had never talked to patients or their families about futile therapy. In the case of half of the respondents (52.11%), there was no such need, while every third respondent (38.42%) did not have a conversation despite indications. Only 6.8% considered themselves sufficiently prepared to hold such a conversation.

Among the most common reasons for undertaking futile therapy in adult patients, the respondents declared (more than one answer was possible) fear of legal liability for not taking such actions (71.58%), as well as fear of being accused of unethical conduct (56.32%), and fear of talking to the patient/patient’s family and their reaction (43.16%). The opinions of the respondents were similar in the context of working with paediatric patients. Detailed data are summarized in Table 3.

### 3.3. Limitation of Futile Therapy and Decisions in This Regard

Most of the respondents (65.26%) believed that the idea of limiting futile therapy was right. At the same time, according to over half of the participants (53.68%), the decision to use or not use futile therapy should not be influenced by economic aspects. Among the indicated reasons for the lack of consent of the patient’s family to limit futile therapy, the lack of acceptance of the inevitability of death prevailed (114 respondents, 60.00%). Every third person (64 respondents, 33.68%) indicated that the reason for the decision was the family’s belief in the supernatural possibilities of medicine. A similar percentage of respondents considered futile therapy as a medical error (31.58%), but what is important, nearly half of the participants (42.11%) did not have an opinion on this subject.

In the opinion of the surveyed nurses, factors facilitating decision-making about limiting futile therapy should include precise eligibility criteria (85.79%), the patient’s declaration of will (78.42%), an unambiguous legal act (76.84%), and educating the public about this issue (74.21%). On the other hand, the most common medical procedures that should be limited or discontinued were extracorporeal membrane oxygenation (ECMO, 62.1%), mechanical and pharmacological support of the circulatory system (61.05%), and renal replacement therapy (58.42%).

An important issue was to indicate the entity that should decide to stop futile therapy. In the case of adult patients, the respondents believed that discontinuation of futile therapy should be decided by the patient (84.21%), followed by a doctor (64.21%). As for paediatric patients, two-thirds of the respondents mentioned a doctor and a court (64.74% and 64.21%, respectively). It is also worth mentioning that the vast majority of staff (72.11%) would agree to withdraw futile treatment if they suffered from an incurable disease. It should be noted that the respondents usually believed (87.37%) that the decision to discontinue futile therapy should be made by the examined person, by a living will, or by a declaration of will. More than half of the respondents (55.79%) would leave this decision to the doctor, and less than half (47.37%) to the patient’s family. Complete results are available in Appendix A.

### 3.4. Determinants of Attitudes and Views of the Respondents

Differentiation of the percentage of opinions on futile therapy in the groups defined by the qualitative variables (gender, place of employment, education, and religiousness, along with the significance of differences between these groups) is available in Appendix A. In relation to any of the important questions in the questionnaire, gender was not statistically significant for the nature of answers provided by the respondents (Appendix A).

In the group of respondents declaring that the comfort of the patient’s last days of life is more important than the need to continue futile therapy, younger respondents prevailed (median age: 43 vs. 45 years, *p* = 0.03). A similar tendency could also be observed among people with shorter work experience, although the difference was not statistically significant (median 14 vs. 24 years, respectively, *p* = 0.13). These observations (differences analysed in the Mann–Whitney *U* test; the results are not included in the table) seem consistent because job seniority is strongly correlated with the age of respondents; younger people may be more willing to accept the limitation of futile therapy in relation to themselves (borderline statistical significance, *p* = 0.08).

Respondents working in ICUs significantly more often than those working in other departments considered themselves sufficiently prepared to talk about futile therapy (*p* = 0.03), assessed this form of therapy as a mistake (*p* = 0.003), and supported the idea of limiting futile therapy (*p* = 0.001). In this group, the opinion that the patient (*p* = 0.001) and the doctor (*p* = 0.0003) should decide about the limitation of futile therapy in adults was more common. As for children, the respondents significantly more often indicated a doctor (*p* = 0.006) and a head of the ward (*p* < 0.0001) as decision-makers. Respondents working in children’s wards significantly more often believed that futile therapy was a mistake (43.59% vs. 23.33%, respectively; *p* = 0.01).

Expressing their will in the form of a declaration was preferred by the vast majority of respondents (no difference depending on the work in the ICU or outside the ICU), while employees of intensive care units would significantly more often entrust the decision to discontinue treatment to a doctor or head of the ward. When it comes to believers, they almost twice as often as non-believers indicated their families as potential decision-makers about futile therapy regarding themselves (50.60% vs. 29.41%, respectively; *p* = 0.05—difference on the border of statistical significance).

## 4. Discussion

The debate on limiting futile therapy in the aspect of End of Life (EoL) care has been going on in Poland over the last decade. The growing demand for EoL care resulting from the aging of societies corresponds to the expectation of a satisfactory quality of life and self-determination. Globalisation, migration, and European integration have a significant impact on society’s changing approach to healthcare. The cultural aspect, value system, and personal experiences have a large impact on the approach to EoL care [8]. Distinguishing the clinical effect from the benefit for the patient underlies the contemporary debate about medical futility, which can be read about in our previous paper [9].

The results of our research indicate that in Poland, represented properly by the population of the Silesian Voivodeship, the continuation of therapy despite the lack of favourable prognosis is still a common practice. According to the surveyed nurses, this is a result of fear of legal liability and fear of being accused of unethical conduct. The concerns of medical personnel are largely determined by legislation because Polish law accepts (but does not regulate) the criteria of futile therapy. In similar studies conducted in the EU countries, doubts have been raised not only by the fact of withdrawing treatment, but rather the determination of the End of Life Decisions (ELD), i.e., who is to decide on withdrawing treatment, defining the role of the family, and maintaining the patient’s autonomy [10,11,12]. In 2012, a meta-analysis of the views and legal situation of EoL care in seven European countries (Belgium, the Netherlands, Norway, Italy, Portugal, Spain, and Germany) was conducted, and the results were published in a separate paper [8]. The cultural context, the professed values, and the model of the doctor–patient relationship (partnership or paternalism) had a significant impact on the approach to the subject in individual countries. There are also differences regarding the role of the patient’s family in end-of-life therapy.

In our study, the respondents who considered themselves religious, almost twice as often indicated their families as the preferred decision-makers to limit futile therapy in relation to themselves.

The Polish Episcopate approved withdrawing futile treatment at the end of life, but this knowledge is still insufficiently propagated in our country [13]. Catholics very often mistakenly treat withdrawing treatment as a behaviour inconsistent with Christian morality. A Swiss study conducted in 2018 indicated that religious medics were less likely to make decisions about withdrawing treatment than those who described themselves as non-believers [14]. The results of this study, as well as the results of a similar study carried out in Belgium [15], indicated that doctors less often decided to withdraw treatment in divorced and single patients due to the lack of a close person who could express the patient’s will. Establishing a formal representative of the patient’s will in a situation when he/she cannot express it, as well as submitting a declaration of will in advance, are currently discussed proposals for the creation of appropriate, binding legal acts in Poland [16].

The analysis of the available literature reveals difficulties in defining the concept of futile therapy [17,18]. Many authors compare this task to the powerlessness of those who attempt to define art or love [19]. In our opinion, the problem stems from the diversity of the patient groups we encounter in everyday clinical practice. The least doubtful are patients who are chronically ill, with no prognosis for survival. Controversies arise around patients in very difficult clinical conditions when there are great doubts as to the satisfactory quality of life after discharge from the ICU [20]. A new category of patients hospitalised in the ICUs, i.e., critically and chronically ill, has also been defined [21]. This group includes patients who survived an acute condition and were hospitalised in the ICU but are not able to live outside this ward. The greatest doubts, however, are raised by paediatric patients [6], which is why our respondents, when asked who should make decisions about withdrawing therapy in children, shifted the burden of responsibility to the courts. Perhaps this is due to the willingness to entrust the decisions to the most objective authority; nevertheless, an unequivocal answer requires in-depth research in this area.

The results of our study show another difficult problem of talking to patients and their families. Both the fear of a conversation and the feeling of insufficient preparation for such a talk underlie the difficulties of the doctor–patient–family relationship. The importance of working out an agreement in an atmosphere of trust is emphasized in most publications on the discussed topic, with a particular emphasis on the need to train young doctors in the ability to conduct such a conversation [22,23]. The role of nurses in establishing a good relationship with the patient’s family is also widely discussed in the literature [17,24]. In Poland, too much importance is attached to the doctor, while reducing the role of the nursing staff. They all agree that the best solution to a conflict situation is to seek consensus and, ultimately, to resort to a court decision [9].

According to the majority of the respondents, the decision to limit futile therapy would be facilitated by the patient’s declaration of will, precise criteria for disqualification from ICU treatment, education, and unambiguous legal acts. The discussion on the “declaration of will” took place in the United States already in the 1960s and is currently taking place in the European Union countries [10,17], Canada [12], and Australia [25]. In Poland, consenting pro future also has many supporters. Leaving in advance a formal document specifying the will to proceed in the event of an unlikely improvement in clinical condition could facilitate the decision-making process and remove responsibility from doctors and the patient’s family. It seems to be the most logical solution, respecting the patient’s freedom and autonomy. At the same time, it should be noted that the vast majority of respondents declared their consent to withdraw futile therapy if the incurable disease concerned them.

It is worth noting that among the people who believed that the comfort of the patient’s last days of life was more important than the continuation of futile therapy, there were significantly more younger nurses, i.e., those with shorter work experience. Similarly, slightly younger respondents would be more likely to accept the limitation of futile therapy in relation to themselves. The reasons for such responses may include a more modern, open-world view of younger people and the affirmation by young people of a full life, not limited by the disease. On the other hand, a more extensive work experience of older employees may result in greater sensitivity, empathy, and understanding of the needs of the patient’s family, and also affect the acceptance of the quality of life in the disease. The analysis of the literature shows that different views of younger and older healthcare workers may also be caused by some changes in the education system, i.e., including ethical issues in the curriculum, and taking up the topic of the rationality of treatment [26,27].

Another element of the discussed problem is the economic aspect. In our study, only every fourth respondent believes that economic considerations should determine the withdrawal of treatment. Considering that the distribution of funds should be fair, with universal access, financial argumentation cannot be completely ruled out. Spending public funds only on continuing therapy that does not bring benefits to the patient raises doubts when at the same time there is a lack of public resources for prophylaxis and quick diagnosis of patients with a prognosis for survival and improvement in the quality of life [28,29].

Our study has several limitations. The strength of the presented study is its innovative character. To the best of our knowledge, the discussed topic, which arouses fear and controversy in society, has not been addressed in our country. Getting to know the opinions of respondents and the determinants of such opinions will help to develop more effective educational programs and may significantly improve the communication process between the medical staff and the patient’s family in a crisis situation such as the decision to withdraw treatment.

The weakness of the study is certainly a significant limitation of inference due to the relatively small size of the study group, i.e., 190 respondents, which is only about 1% of all professionally active nurses registered in the Silesian Voivodeship in 2020. The obtained results should be treated with great caution, but it is worth noting that they contribute to an in-depth analysis that should be carried out on a larger group of nursing staff.

Summing up, it is worth emphasizing that the respondents treated this issue as really important and devoted their valuable time to filling in the questionnaire. The vast majority of the respondents believe and agree that the comfort of the patient’s last days is more important than the persistent continuation of therapy and prolonging life at all costs. The presented results clearly show the attitude of the respondents who defend the patient’s dignity and autonomy. In Anglo-Saxon literature, this approach is referred to as dignity-conserving care [22]. The analysis of the collected results allows the conclusion that the problem of futile therapy is familiar to nurses. Undoubtedly, further education is needed, both among medical and future patients. Conversations about death are still taboo in our country and belief in the power of medicine is huge. Polish society needs a wide-ranging debate on end-of-life care and the futility of therapy. Hope rests on young healthcare professionals, who are still untainted by paternalism, who speak patiently with patients, and who value autonomy and freedom above all else. It is essential to educate this group of healthcare workers, as they will guarantee dignified conditions in the last moments of a patient’s life, and also in intensive care units. Therefore, in our opinion, it is necessary to introduce an additional subject at medical universities that would combine elements of ethics, law, intensive care, and end-of-life therapy. It should be emphasized that a patient with limited futile therapy is taken over by palliative care. Therefore, it should be explained that such a patient is not abandoned by medicine, and the role of the nursing staff in palliative care is crucial. Knowledge of the subject in this professional group is necessary to accept the limitation of futile therapy in our country. Another field of activity is legislation; the legal acts in force will allow us to reduce and possibly eliminate the concerns arising from the lack of appropriate legislation.

## 5. Conclusions

The issue of futile therapy and related clinical implications is known to the nursing community. Knowledge of the problem is passive. Lack of education, positive examples from doctors managing departments, fear of legal liability, as well as ethical and ideological doubts result in the fact that futile therapy is still commonly implemented in healthcare facilities.

Education and public debate on end-of-life therapy have a better chance of making a difference in hospital wards because waiting for a clear legal act regulating the principles of withdrawing from futile therapy will postpone the solution of the problem for the next years.

## Figures and Tables

**Table 1 ijerph-19-16975-t001:** General characteristics of the studied population, including basic qualitative variables.

Variable	Response Category	*n*	%	Variable	Response Category	*n*	%
Education	Medical high school	25	13.16	Type of hospital/ward *	Teaching hospital with ICU	117	61.58
	Medical vocational school	5	2.63		Non-teaching hospital with ICU	38	20.00
	Bachelor of Nursing	112	58.95		Hospital without ICU	10	5.26
	Master of Nursing	42	22.11		Ward with Intensive Care Unit	16	8.42
	nd	6	3.16		Ward without Intensive Care Unit	3	1.58
Work with children	Yes	78	41.05		Another place	12	6.32
	No	90	47.37	Religion	Catholic	114	60.00
	nd	22	11.58		Practising Catholic	55	28.95
Work in the ICU	Yes	120	63.16		Other	4	2.11
	No	64	33.68		Atheist	17	8.95
	nd	6	3.16				

Min—minimum; Max—maximum; IQR—interquartile range; *n*—number; ICU—intensive care unit nd—no data; *—multiple choice question.

**Table 2 ijerph-19-16975-t002:** Work experience and employees’ views on futile therapy (numbers and percentages in brackets).

Contact with Dying Patients at Work				
Variable	Probably not	Several times a year	Quite often	nd
Does the specificity of professional work involve contact with dying patients?	30 (15.79%)	43 (22.63%)	115 (60.53%)	2 (1.05%)
**Medical Procedures That, in the Opinion of the Respondents, Should Be Limited during Futile Therapy**				
Variable	Yes	No opinion	No	nd
Patient intubation and mechanical ventilation	84 (44.21%)	49 (25.79%)	56 (29.47%)	1 (0.53%)
Mechanical and pharmacological support of the circulatory system	116 (61.05%)	43 (22.63%)	28 (14.74%)	3 (1.58%)
Extracorporeal membrane oxygenation (ECMO)	118 (62.10%)	49 (27.79%)	22 (11.58%)	1 (0.53%)
Renal replacement therapy	111 (58.42%)	47 (24.74%)	29 (15.26%)	3 (1.58%)
Transfusion of blood products	101 (53.16%)	53 (27.89%)	35 (18.42%)	1 (0.53%)
Antibiotic therapy	77 (40.53%)	55 (28.95%)	57 (30.00%)	1 (0.53%)
**Professional Contact with Futile Therapy**				
Variable	Yes	No	-	nd
Contact with futile therapy for adults in professional practice	142 (74.74%)	39 (20.52%)	-	9 (4.74%)
Contact with futile therapy for children in professional practice	57 (30.00%)	110 (57.89%)	-	23 (12.10%)
**Opinions of Respondents on the Necessity of Futile Therapy**				
Variable	Yes	Depends on the patient’s situation	No	-
Use of a futile therapy protocol in the adult ICU	66 (37.74%)	1 (0.53%)	122 (64.21%)	-
Use of futile therapy despite the lack of prognosis for improvement	72 (37.89%)	66 (34.74%)	52 (27.37%)	-
**Experience in Conducting a Conversation about Futile Therapy**				
Variable	Yes	No, despite the need	No, there was no need	-
Talking to family/patient about futile therapy	18 (9.47%)	73 (38.42%)	99 (52.11%)	-
**Opinions of the Respondents**				
Variable	Yes	No opinion	No	nd
Prepared to talk to the patient about futile therapy	13 (6.84%)	55 (28.95%)	122 (64.21%)	-
Use of futile therapy is a mistake	60 (31.58%)	80 (42.11%)	50 (26.32%)	-
Idea of limiting futile therapy is right	124 (65.26%)	52 (27.37%)	13 (6.84%)	1 (0.53%)
Economic aspects should be decisive in stopping futile therapy	47 (24.74%)	41 (21.58%)	102 (53.68%)	-

nd—no data.

**Table 3 ijerph-19-16975-t003:** Reasons for starting futile therapy in the population of paediatric and adult patients.

Declared Reason for Making Decisions about Starting Futile Therapy	In Children		In Adults	
	*n*	%	*n*	%
Fear of talking to the patient/patient’s family and their reaction	110	57.89	82	43.16
Fear of legal liability for withdrawing or withholding treatment	135	71.05	136	71.58
Heroic fight for life to the end	84	44.21	50	26.32
Fear of being accused by colleagues of a lack of professional ethics	35	18.42	24	12.63
Fear of being accused by the patient’s family of a lack of professional ethics	102	53.68	107	56.32
Order/recommendation from the supervisor	50	26.32	56	29.47
Passivity	14	7.37	21	11.05

*n*—number, %—percentage.

## Data Availability

The data are available on request from the Department of Epidemiology, Medical University of Silesia in Katowice. The request should be formulated and sent to epikat@sum.edu.pl.

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
