# Peer review of "Limitation of Futile Therapy in the Opinion of Nursing Staff Employed in Polish Hospitals—Results of a Cross-Sectional Study"

_ijerph, 2022, doi:10.3390/ijerph192416975_

Round 1
Reviewer 1 Report
Thank you for submitting your article to the Journal.
Abstract
line 12 Should this read … an original questionnaire … (or “a newly designed questionnaire” to make it clear)
Line 13 …. 190 nursing …. should be 190 nurses …
Also, on line 44 you say nursing and medical staff and on line 47 you say medical staff.
This needs clarifying
Line 21 do you have a percentage for “vast majority”
Introduction
You refer to Surviving Sepsis Campaign guidelines but I think this needs putting in context for readers unfamiliar with these guides. For example, are you implying that treatment of multiorgan failure after or during sepsis is futile?
Line 44 see comment above about “original questionnaire”
Line 67 gratification better to say gratuity.
Line 69 again you say nurses see earlier comments
Line 90 does “… weekly participation in the service”. Refer to working “part time” or attending the Catholic church? Please clarify.
Discussion
Line 192 should this read … in our earlier paper (8). ??
Line 301 better to say medical or health care staff than “medics”
You have a very experienced workforce encountering death on a “regular” basis and with access to futile therapy protocols but only a low number feeling sufficiently prepared to discuss these issues.
You note “supernatural possibilities” or unrealistic expectations from the family.
You discuss the legal liability and unethical conduct as reasons for continuing futile therapy.
You discuss the EoL experience in the EU, the role of the Catholic church for believers, and the added challenges in the paediatric population.
Community attitudes possibly changing between younger healthcare professionals and older or more senior ones
Given the above your Summary could be more definitive.
Consider recommending or identifying the need for community and health care professional education and or debate and how that might be facilitated.
Community forums at village or town level possibly promoted by politicians(???) or respected community figures and similar debate/education in medical and nursing undergraduate courses.
Perhaps changing the laws are more complex and difficult to achieve?
Reviewer 2 Report
Firstly, congratulate you on your article, it has positive things, however I wanted to make some observations.
In table 5 the values ​​of p some present a decimal others more than one. You should try to put at least 2 decimal places. to homogenize except when the values ​​are significant. In relation to methods, the description of the questions in the questionnaire does not refer to whether the questionnaire is validated or not validated, if it was reviewed by experts, or if an initial pilot test was carried out, also within the limitations of being 18 questions out of 28. there are confounding factors such as response bias as it is such an extensive questionnaire. I think that the conclusions are not so systematically thought about why, giving interpretation to what the obtained results want to represent.
Round 2
Reviewer 2 Report
I have just finished reviewing your document and I am glad to find new things, or that failing that cannot be seen in the previous version, for that reason, in context, I would recommend being able to validate your scale. CongratulationsAuthor Response
Please see the attachment
